# Efficient Minimum Bayes Risk Decoding using Low-Rank Matrix Completion Algorithms

**Firas Trabelsi**
Google
firast@google.com

**David Vilar**
Google
vilar@google.com

**Mara Finkelstein**
Google
marafin@google.com

**Markus Freitag**
Google
freitag@google.com

## Abstract

Minimum Bayes Risk (MBR) decoding is a powerful decoding strategy widely used for text generation tasks, but its quadratic computational complexity limits its practical application. This paper presents a novel approach for approximating MBR decoding using matrix completion techniques, focusing on the task of machine translation. We formulate MBR decoding as a matrix completion problem, where the utility metric scores between candidate hypotheses and pseudo-reference translations form a low-rank matrix. First, we empirically show that the scores matrices indeed have a low-rank structure. Then, we exploit this by only computing a random subset of the scores and efficiently recover the missing entries in the matrix by applying the Alternating Least Squares (ALS) algorithm, thereby enabling a fast approximation of the MBR decoding process. Our experimental results on machine translation tasks demonstrate that the proposed method requires 1/16 utility metric computations compared to vanilla MBR decoding while achieving equal translation quality measured by COMET22 on the WMT22 dataset (en ↔de, en ↔ru). We also benchmark our method against other approximation methods and we show gains in quality when comparing to them.

## 1 Introduction

The generation process in most conditional natural language processing tasks is usually guided by the maximum-a-posteriori (MAP) rule: given an input $x$, generate the output $\hat{y}$ that maximizes the posterior probability distribution: $\hat{y} = \operatorname{argmax}_y p(y|x)$. It can be shown that MAP decoding is optimal under a 0-1 loss criterion. However for more nuanced tasks, where different outputs can be considered correct, and the quality of the output is not just a binary decision between "right" and "wrong", MAP decoding is no longer optimal. Neural Machine Translation (NMT) is a prominent example of these types of tasks. For NMT, a system is trained to generate a sentence in a target language given a source sentence in another language. For a given sentence, there exists a variety of possible translations, each of which has a different quality level. Eikema and Aziz (2020) demonstrated that MAP decoding methods are suboptimal for NMT, and showed that other generation strategies may be preferred. Furthermore, NMT models often assign human translations lower probabilities than their own beam search outputs, due to model calibration issues (Ott et al., 2018; Freitag et al., 2020).

As an alternative, Eikema and Aziz (2020, 2022) applied MBR decoding for NMT models. MBR decoding follows a two-step approach, where a model is used to generate a list of candidate translations and a list of pseudo-references (which may be the same as the list of candidates). The candidates

38th Conference on Neural Information Processing Systems (NeurIPS 2024).

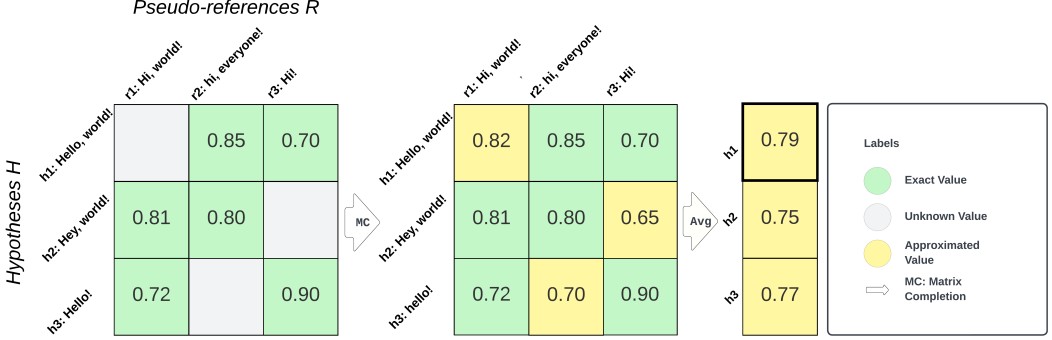

Figure 1: **PMBR decoding only requires a subset of the utility computations to approximate the output of MBR decoding.** The method approximates the unknown values by running a matrix completion algorithm which exploits the low-rank nature of the MBR matrix. Once the full matrix is recovered, the method behaves similar to the vanilla MBR decoding method where the hypothesis with the highest average score is selected.

are then scored with a performance metric using the pseudo-references as an approximation of the true references, and the candidate with the maximum expected quality (or equivalently minimum risk) is then selected. In contrast to MAP decoding, MBR decoding is not designed to generate the translation with the highest estimated model probability; instead it aims to directly optimize a utility function. Subsequent research conducted by Freitag et al. (2022a) showed that MBR decoding with *neural* utility metrics leads to significant improvements over beam search decoding. However, MBR is computationally expensive, with a time complexity of $O(N^2)$ for a candidate list containing $N$ samples and $N$ pseudo-references (usually the two lists coincide). According to Freitag et al. (2022a), ideally $N$ ranges between $100$ and $1\,000$, which involves thousands to millions of utility function computations. Note than when using neural metrics, each of the $O(N^2)$ "computation steps" is itself expensive, requiring a forward pass through a large neural network.

In this work, we propose to reduce the number of metric computations by scoring only a subset of candidate–pseudo-reference pairs. We then proceed to use a matrix completion algorithm (ALS in our case) to estimate the remaining utility scores. For such completion algorithms to work, the full matrix has to fulfill some conditions, specifically to be a low-rank matrix. We empirically show that this is indeed the case for the utility matrices for MBR decoding. Intuitively, this makes sense: akin to recommendation systems, where similar users are expected to have similar book or movie preferences, similar pseudo-references are expected to have similar "translation preferences". Experimental results show the effectiveness of our method, where the performance of the full MBR algorithm can be matched with only a fraction of the computational cost. Compared to other approaches that also seek to reduce the number of computations, our method does not compromise translation quality, as confirmed by human evaluation.

Our scientific contributions are as follows:

1. We empirically show that the utility matrices for MBR decoding are low-rank.
2. We apply ALS to a subset of scores to approximate the full MBR matrix.
3. We show that using our method we can reduce the number of computations by a factor of 16, while maintaining the same translation quality level.

## 2   Related Work

While MT research has traditionally relied on MAP decoding or generating $k$-best lists through beam search for MBR decoding, Eikema and Aziz (2020) proposed an approximation of MBR decoding via unbiased sampling. Their method aims to address the limitations of MAP decoding (Eikema and Aziz, 2020; Müller and Sennrich, 2021; Eikema and Aziz, 2022) by demonstrating that samples drawn from the NMT model align more faithfully with training data statistics when compared to beam

search. Freitag et al. (2022a) showed that using neural metrics results in significant improvements in translation quality. Freitag et al. (2023a) reported that the choice of sampling approach is important, and epsilon sampling (Hewitt et al., 2022) is ideal for MBR decoding and reranking.

While the improvements in translation quality afforded by MBR are widely acknowledged, its high computational cost limits its application in practice. Different approaches have been proposed to speed up MBR computation. Eikema and Aziz (2022) propose to decouple the candidate and pseudo-reference lists to allow for different sizes, and propose a coarse-to-fine refinement of the hypothesis space. Cheng and Vlachos (2023) speed up MBR decoding by gradually increasing the number of samples used to estimate the utility, while additionally pruning the hypothesis space. Jinnai and Ariu (2024) formulate MBR as a medoid identification problem, and apply approximate algorithms developed on this problem. Vamvas and Sennrich (2024) aggregate the set of pseudo-references, allowing for just one utility computation per candidate. This greatly accelerates the decoding process, but the utility metric needs to fulfill certain conditions to be applicable. Finkelstein et al. (2024) use MBR decoding in a knowledge-distillation framework to simulate MBR decoding with single-pass search. Tomani et al. (2024) train quality-aware translation models in order to reduce the size of the candidate list. Similar in spirit to MBR decoding, QE-rescoring approaches (Fernandes et al., 2022) also directly optimize a utility function, with linear-time cost.

Low-Rank Matrix completion is an active area of research and multiple algorithms have been developed to perform it. Nguyen et al. (2019) is an extensive survey for such methods. Some of the popular algorithms are: Singular Value Thresholding (Cai et al., 2008), Bayesian Probabilistic Matrix Factorization (Akulwar and Pardeshi, 2016), Maximum Margin Matrix Factorization (Srebro et al., 2004) and Alternating Least Squares (Zachariah et al., 2012), which is the one we chose for this work. To the best of our knowledge, this work is the first one to apply matrix completion algorithms for completing a partial MBR score matrix.

# 3 Preliminaries

We are given an NMT model $P_\Theta(y|x)$ which serves to estimate the probability of a hypothesis segment $y$, given a source segment $x$, with $\Theta$ being the learned parameters of the neural network. MAP decoding involves searching for the most probable translation under $P_\Theta(y|x)$. However, determining the hypothesis with the maximum probability is computationally intractable due to the expansive and combinatorially complex search space. Consequently, approximations like beam search (Graves, 2012; Sutskever et al., 2014) are often employed.

If we want to generate diverse hypotheses, e.g. in generative tasks where creativity is desired instead of selecting the candidate with the highest probability (or an approximation thereof), we sample the output sentence following the probability distribution defined by the model. For NMT, this approach is used for generating a list of candidate translations. Specifically, epsilon sampling, as outlined by Hewitt et al. (2022), has emerged as the leading sampling technique for MBR. It was shown by Freitag et al. (2023a) to outperform other methods such as ancestral, top-$k$ or nucleus sampling (Holtzman et al., 2020). Epsilon sampling prunes away any token with a probability lower than a threshold $\varepsilon$, thereby guaranteeing that each token within a sample is allocated a fair probability mass.

## 3.1 Minimum Bayes Risk Decoding

In MBR decoding (Bickel and Doksum, 1977; Berger, 1985), given a set of candidate hypotheses $\mathcal{H}$, the goal is to select the optimal hypothesis based on its expected utility, measured by a function $u$, with respect to the distribution over human references within the space of all references $\mathcal{Y}$.

Since the true distribution remains unknown, we resort to sampling from the model instead, which relies on the assumption that the model provides a reliable approximation for the true underlying distribution over human translations. Furthermore, the integration over the vast space of all possible references $\mathcal{Y}$ is computationally intractable. Therefore, MBR adopts a finite sample estimate by sampling a set of pseudo-references $\mathcal{R}$ from $P_\theta(\cdot|x)$. This approximation can be expressed as:

$$h^{\text{MBR}} = \underset{h \in \mathcal{H}}{\operatorname{argmax}} \frac{1}{|\mathcal{R}|} \sum_{r \in \mathcal{R}} u(h, r) \tag{1}$$

Usual practice is to set $\mathcal{H} = \mathcal{R}$, i.e. the same set of model hypotheses serves both as the candidate list $\mathcal{H}$ as well as the pseudo-reference list $\mathcal{R}$. The computational time complexity of MBR decoding is $O(N^2)$ with $N$ the size of the candidate list.

Note that this quadratic expression refers to *each sentence* to translate, i.e. for a corpus of size $S$, the total cost will be $O(S \cdot N^2)$. Also there is a hidden (multiplicative) constant, namely the cost of the computation of the utility function. For surface level metrics (e.g. BLEU, ChrF), this cost is negligible, but for neural metrics it involves computing the forward pass of a large neural network. Therefore, any reduction in the number of metric computations has an important effect on the total running cost.

## 3.2 Low-Rank Matrix Completion

Low-Rank Matrix Completion is a fundamental problem in machine learning and data analysis with popular application such as *Collaborative Filtering* (Rennie and Srebro, 2005) and *Image Denoising* (Candes and Recht, 2008). The goal of matrix completion is to estimate the missing entries of a partially observed matrix, under the assumption that the underlying matrix is low-rank. This assumption implies that the matrix can be well-approximated by a product of two smaller matrices, capturing the latent factors that explain the observed data. Candes and Recht (2008) proved that perfect approximations can be achieved if the number of observed entries is larger than $CN^{1.2}r\log(N)$ for some positive numerical constant $C$, for most $N \times N$ matrices of rank $r$ with very high probability.

One simple and efficient algorithm is Alternating Least Squares (ALS) (Zachariah et al., 2012). To recover any matrix $M$, the algorithm approximates it by two smaller matrices $M \approx X^T Y$ and then minimizes the following equation given the observed entries.

$$\min_{X,Y} \sum_{m_{ij} \text{ observed}} (m_{ij} - x_i^T y_j)^2 + \lambda \left( \sum_i ||x_i||^2 + \sum_j ||y_j||^2 \right) \tag{2}$$

The algorithm achieves this by alternatively solving for $X$ and $Y$ as shown in Algorithm 1 The algorithm has three hyperparameters: $\lambda$ a regularization term, $r$ the second dimension of the smaller matrices and $n$ the number of alternating steps performed. The main motivation for picking this algorithm in our approach is its simple implementation.

---

**Algorithm 1** ALS for Matrix Completion

---

**Require:** $\lambda$, $r$ and $N$
1: Initialize $X, Y$ with shapes $N \times r$ and $r \times N$
2: **repeat**
3:     **for** $i = 1 \ldots n$ **do**
4:         $x_i = \left( \sum_{m_{ij} \in m_{i*}} y_j y_j^\top + \lambda I_k \right)^{-1} \sum_{m_{ij} \in m_{i*}} m_{ij} y_r$
5:     **end for**
6:     **for** $j = 1 \ldots n$ **do**
7:         $y_j = \left( \sum_{m_{ij} \in m_{*j}} x_r x_r^\top + \lambda I_k \right)^{-1} \sum_{m_{ij} \in m_{*j}} m_{ij} x_i$
8:     **end for**
9: **until** convergence

---

# 4 MBR Matrix

## 4.1 Definition of MBR matrix

Given a source sentence, we use an NMT model to generate a set $\mathcal{H}$ of hypotheses such that $|\mathcal{H}| = N$. As explained in the preliminaries section, the MBR method uses two different sets of hypotheses and pseudo-references, but in practice we use the same set of samples for both $\mathcal{H}$ and $\mathcal{R}$. The pairwise scores for all hypotheses in $\mathcal{H}$ gives an $N \times N$ matrix $M$ such that $M[i, j] = U(h_i, h_j)$ for all $(h_i, h_j) \in \mathcal{H} \times \mathcal{H}$ and a utility metric $U$ that computes some similarity between two hypotheses.

Table 1: Summary of the first three singular values of MBR matrices for the MetricX and chrF utility functions, with two different sizes and four different language pairs

| | 64x64 | | | | | | 128x128 | | | | | |
| | MetricX | | | chrF | | | MetricX | | | chrF | | |
| LP | $\sigma_1$ | $\sigma_2$ | $\sigma_3$ | $\sigma_1$ | $\sigma_2$ | $\sigma_3$ | $\sigma_1$ | $\sigma_2$ | $\sigma_3$ | $\sigma_1$ | $\sigma_2$ | $\sigma_3$ |
|---|---|---|---|---|---|---|---|---|---|---|---|---|
| English→German | 45.7 | 2.1 | 1.0 | 39.9 | 2.4 | 1.4 | 91.6 | 3.8 | 1.7 | 76.2 | 4.2 | 2.1 |
| German→English | 47.4 | 2.1 | 1.4 | 47.5 | 1.5 | 1.4 | 94.6 | 3.7 | 2.1 | 93.7 | 3.0 | 2.0 |
| English→Russian | 47.7 | 2.0 | 1.1 | 36.1 | 1.9 | 1.3 | 95.4 | 3.6 | 1.7 | 76.8 | 3.4 | 2.1 |
| Russian→English | 46.1 | 2.2 | 1.0 | 40.5 | 2.8 | 1.3 | 92.0 | 4.1 | 1.7 | 81.8 | 5.7 | 2.1 |
| Average | 46.7 | 2.1 | 1.1 | 41.0 | 2.15 | 1.35 | 93.4 | 3.8 | 1.8 | 82.1 | 4.1 | 2.1 |

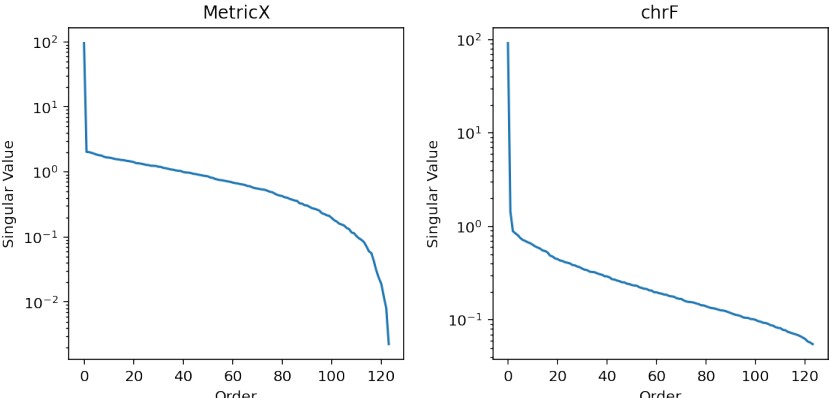

Figure 2: Plot the singular values of an example 124x124 MBR matrix using logscale. We observe a sharp drop after the first singular value for the two utility metrics indicating that the matrix is rank-1.

With this matrix formulation, MBR decoding reduces to picking the row with the highest average (since each row maps to one sample in the hypotheses list).

### 4.2 MBR matrices are low rank

Intuitively, the values within the MBR matrix are highly correlated, since by definition each value $M[i, j]$ is computing a similarity score between two hypotheses given a utility metric. This is a key assumption that our work is built on since low-rank matrices have theoretical bounds on the number of entries required to recover the full matrix (Candes and Recht, 2008).

We verify this assumption empirically. We generated 1024 samples for each example in the WMT 2022 en ↔de and en ↔ru datasets. We then generated the $N \times N$ matrices for different values of $N \in \{64, 128\}$ by only considering a random subset of the samples using two different utility metrics: MetricX (Freitag et al., 2022b) and ChrF (Popović, 2015). We then perform singular value decomposition and look at the distribution of the singular values, shown in Figure 2 and Table 1. We observe that across utility metrics and matrix dimensions, $\sigma_1 \gg \sigma_2$. On average across datasets we have $\sigma_2/\sigma_1 < 0.05$, this means that most of the information within the matrix can be captured by a single dominant direction or component and thus can be approximated by a rank-1 matrix.

## 5 The PMBR Method

We propose an approximation method for MBR decoding that leverages the low-rank structure of the MBR matrix. The procedure is shown in Algorithm 2. Given an NMT model, we start by generating a set of hypotheses $\mathcal{H}$ similar to the vanilla MBR method. Then, instead of computing all the pairwise scores in the utility matrix, we only compute a random subset of the scores that we denote with $\Omega$. The size of $\Omega$ depends on the *computation budget* available. We define the budget as the ratio of

computations performed with respect to the total amount of computations to compute the full matrix. Thus for any given budget $1/r$, we end up with $N^2/r$ entries observed in the matrix. The next step is to apply ALS on $M[\Omega]$ as described in Algorithm 1, where $M[\Omega]$ denotes the matrix of size $N \times N$ where only the entries of $\Omega$ are non-null. Finally, with all the pairwise scores recovered, we perform vanilla MBR decoding. We call this procedure PMBR for *Probabilistic MBR* decoding.

---

**Algorithm 2** PMBR: MBR Approximation using ALS

---

**Require:** List of hypotheses $H$, reduction ratio $r \in (0, 1)$
 1: $N \leftarrow |H|$
 2: $S \leftarrow \lceil |N|^2 \cdot r \rceil$          ▷ Number of utility computations
 3: $\Omega \leftarrow$ Sample $S$ coordinate pairs $(i, j)$ from $N \times N$
 4: $M \leftarrow \mathbf{0}^{|N| \times |N|}$           ▷ Initialize empty matrix
 5: **for** $(i, j) \in \Omega$ **do**
 6:   $M_{ij} \leftarrow U(i, j)$
 7: **end for**
 8: $M \leftarrow \text{ALS}(M, \text{hyperparameters})$
 9: **return** vanilla_MBR(M)

---

**Time Complexity**   The time complexity of this algorithm is dominated by the utility metrics computations. The utility metrics are deep neural networks that require a number of floating-point operations in the order of millions while ALS requires only a few hundred operations. For reference, 30 steps of the ALS algorithm with $r = 10$ running on a CPU takes on average 0.2 seconds to run while the MetricX inference takes 3.4 seconds on a TPUv4 platform. Thus, the savings in run time achieved by our approximation is close to proportional to the savings in number of utility computations. Note that this analysis focuses only on the second stage of MBR decoding, i.e. we do not take the cost of generating the hypotheses into account.

# 6   Experimental Setup

## 6.1   Metrics

We use MetricX (Juraska et al., 2023) as the utility function for all variants of MBR decoding as it has been shown that neural fine-tuned metrics outperform word-overlap metrics like BLEU (Papineni et al., 2002) and ChrF (Popović, 2015) for MBR decoding (Freitag et al., 2022a). MetricX is an extension of BLEURT (Sellam et al., 2020), showing higher correlation with human judgment (Freitag et al., 2023b) and has been designed to also work on multi-sentence segments (Deutsch et al., 2023) and not only sentences in isolation. In addition, we report COMET22 (Rei et al., 2020, 2022) scores as there is a risk of overfitting (Amrhein and Sennrich, 2022) on MetricX. In addition, for one selected experiment we conducted expert-based human evaluations using MQM (Freitag et al., 2021), a human evaluation scheme centered on marking errors present in the translations. We report results by varying the budget available to the MBR methods. For each budget, we randomly sample from the full MBR matrix, and report the average results of 1000 trials.

## 6.2   Datasets and Model

We run experiments using the WMT 2022 test sets for English↔German (en↔de) and English↔Russian (en↔ru). The official WMT test sets (Kocmi et al., 2022) are split into sentences but come with document information. We constructed multi-sentence (paragraph) level test sets with the following method: for each document, we concatenate sentences together as long as we do not exceed 500 sentence piece model (SPM) tokens (given the MetricX SPM model). We respect sentence boundaries and do not truncate sentences. In WMT22, there are four different domains. Some domains lack document context, so segments remain as single sentences, even within multi-sentence test sets. Test data statistics can be seen in Appendix 5. We use PaLM8B (Chowdhery et al., 2022) as translation model and sample 1024 examples for each sentence using epsilon sampling with $\epsilon = 0.02$ (Freitag et al., 2023a) and using 3-shot prompting with examples taken from the FLORES corpus (Guzmán et al., 2019)

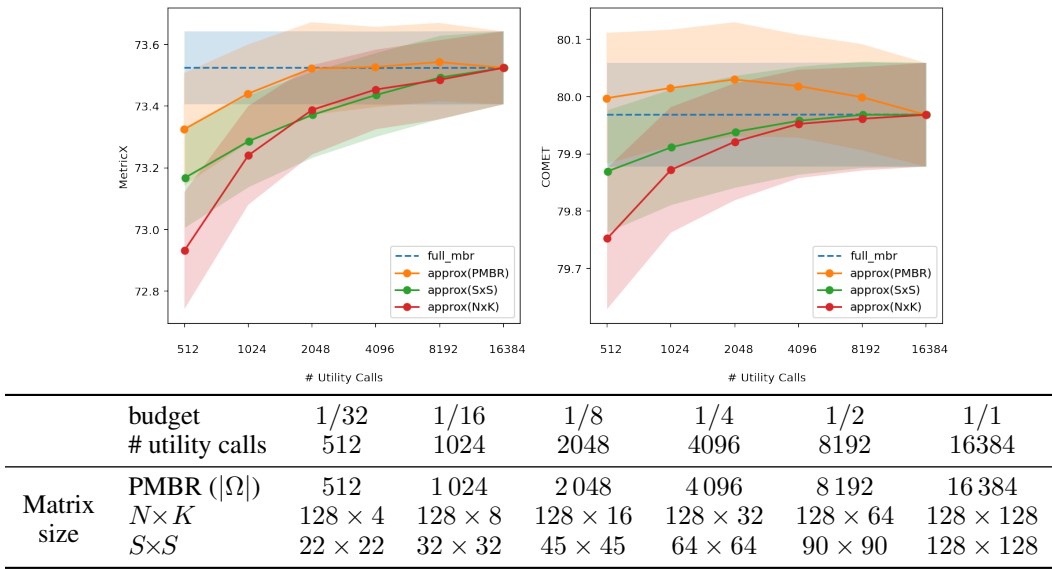

| | budget | 1/32 | 1/16 | 1/8 | 1/4 | 1/2 | 1/1 |
|---|---|---|---|---|---|---|---|
| | # utility calls | 512 | 1024 | 2048 | 4096 | 8192 | 16384 |
| Matrix size | PMBR ($|\Omega|$) | 512 | 1 024 | 2 048 | 4 096 | 8 192 | 16 384 |
| | $N \times K$ | $128 \times 4$ | $128 \times 8$ | $128 \times 16$ | $128 \times 32$ | $128 \times 64$ | $128 \times 128$ |
| | $S \times S$ | $22 \times 22$ | $32 \times 32$ | $45 \times 45$ | $64 \times 64$ | $90 \times 90$ | $128 \times 128$ |

Figure 3: We scored WMT22 DeEn dataset 1000 times for each budget available. Each scoring picks without replacement 128 samples from the 1024 samples available for each sentence. The highlighted area shows the standard deviation of the scores.

## 6.3 Decoding Methods

We compare our approximation PMBR against three other decoding methods. To enable a fair comparison, we adapt each method so that the number of utility function computations is the same for each method. (Recall that for a given budget $1/r$, we only observe $N^2/r$ entries in the matrix when performing PMBR.) We compare PMBR with the following methods:

- FMBR: This is the full MBR method. This is the only method that is not affected by the budget i.e the full matrix is observed.
- $N \times K$: This method was proposed by Eikema and Aziz (2022) and works by shrinking the pseudo-references list size. For a budget $1/r$ the MBR matrix gets reduced to an $N \times K$ matrix with $K = N/r$. The $K$ pseudo-references are randomly sampled.
- $S \times S$: This method corresponds to FMBR, but reduces the total size of the utility matrix to a size of $S \cdot S$, where the total number of entries corresponds to the available budget, i.e. $S = \sqrt{N^2/r}$. The $S$ examples are randomly sampled.

## 6.4 Hyperparameter Tuning

The ALS algorithm has three hyperparameters $\lambda$, $n$ and $r$ as described in Algorithm 1. We perform a grid search to optimize these hyperparameters, setting our loss function to be the accuracy with respect to the vanilla MBR method. Concretely, for each example sentence we rank all samples by running the vanilla MBR. Let us denote with $h_{\text{MBR}}$ the hypothesis selected by the full MBR method, and let $\text{pos}_{\text{PMBR}}(h_{\text{MBR}})$ the rank the position of $h_{\text{MBR}}$ after ordering the hypotheses according to the scores predicted by PMBR. The loss function is then just the sum of $\text{pos}_{\text{PMBR}}(h_{\text{MBR}})$ for all the hypotheses in a subset of the data. We minimize this loss per language pair on 10 examples that we hold from the data generated with the WMT 2022 datasets with this search space $\{\lambda \in \{0.1, 0.15, 0.2\}\} \times \{r \in \{5, 6, \ldots 14, 15\}\} \times \{n \in \{10, 11, \ldots, 29, 30\}\}$

## 7 Results

The main experimental results are summarized in Table 2 and Table 3. In Table 2, we fix $N = 128$ and we study the behaviour of each approximation method by limiting their budget to a fraction of the

Table 2: Results on the four translation directions on the WMT22 data. Each number (except for FMBR) is the average of 1 000 runs with different random values taken from the full MBR matrix. $N$ is set to 128, and the budget is allocated according to the description in Section 6.3. 'C' denotes COMET22 scores and 'X' MetricX scores.

| | budget | 1/32 | | 1/16 | | 1/8 | | 1/4 | | 1/2 | |
|---|---|---|---|---|---|---|---|---|---|---|---|
| | | C | X | C | X | C | X | C | X | C | X |
| en→de | FMBR | 83.52 | 77.01 | 83.52 | 77.01 | 83.52 | 77.02 | 83.52 | 77.02 | 83.52 | 77.01 |
| | PMBR | **83.53** | 75.94 | **83.63** | **76.50** | **83.63** | **76.81** | **83.60** | **76.96** | **83.56** | **77.01** |
| | $N{\times}K$ | 82.18 | 74.96 | 82.90 | 75.99 | 83.28 | 76.59 | 83.45 | 76.84 | 83.48 | 76.93 |
| | $S{\times}S$ | 83.39 | **76.12** | 83.52 | 76.43 | 83.57 | 76.63 | 83.59 | 76.79 | 83.57 | 76.91 |
| de→en | FMBR | 79.97 | 73.52 | 79.97 | 73.52 | 79.97 | 73.52 | 79.97 | 73.52 | 79.97 | 73.52 |
| | PMBR | **80.00** | **73.32** | **80.01** | **73.44** | **80.03** | **73.52** | **80.02** | **73.53** | **80.00** | **73.54** |
| | $N{\times}K$ | 79.75 | 72.93 | 79.87 | 73.24 | 79.92 | 73.39 | 79.95 | 73.45 | 79.96 | 73.49 |
| | $S{\times}S$ | 79.87 | 73.17 | 79.91 | 73.29 | 79.94 | 73.37 | 79.96 | 73.44 | 79.97 | 73.49 |
| en→ru | FMBR | 83.52 | 77.01 | 83.52 | 77.01 | 83.52 | 77.02 | 83.52 | 77.02 | 83.52 | 77.01 |
| | PMBR | **83.53** | 75.94 | **83.63** | **76.50** | **83.63** | **76.81** | **83.60** | **76.96** | **83.56** | **77.01** |
| | $N{\times}K$ | 82.18 | 74.96 | 82.90 | 75.99 | 83.28 | 76.59 | 83.45 | 76.84 | 83.48 | 76.93 |
| | $S{\times}S$ | 83.39 | **76.12** | 83.52 | 76.43 | 83.57 | 76.63 | 83.59 | 76.79 | 83.57 | 76.91 |
| ru→en | FMBR | 79.17 | 75.57 | 79.17 | 75.57 | 79.17 | 75.57 | 79.17 | 75.57 | 79.17 | 75.57 |
| | PMBR | **79.15** | **75.15** | **79.23** | **75.40** | **79.22** | **75.48** | **79.19** | 75.51 | **79.18** | **75.56** |
| | $N{\times}K$ | 78.71 | 74.68 | 78.99 | 75.21 | 79.08 | 75.41 | 79.14 | **75.52** | 79.15 | 75.54 |
| | $S{\times}S$ | 78.98 | 75.01 | 79.06 | 75.20 | 79.10 | 75.34 | 79.13 | 75.43 | 79.15 | 75.52 |

Table 3: Summary of the average scores of the full DeEn WMT 2022 pairs scored 1000 times using MetricX and COMET22 as evaluation metrics while varying the size of the hypothesis list

| | budget | 1/32 | | 1/16 | | 1/8 | | 1/4 | | 1/2 | |
|---|---|---|---|---|---|---|---|---|---|---|---|
| | | C | X | C | X | C | X | C | X | C | X |
| N=32 | FMBR | NA | NA | 79.91 | 73.29 | 79.91 | 73.29 | 79.91 | 73.29 | 79.91 | 73.29 |
| | PMBR | NA | NA | **79.87** | **73.01** | **79.97** | **73.26** | **79.99** | **73.36** | **79.95** | **73.34** |
| | NxK | NA | NA | 79.46 | 72.27 | 79.74 | 72.84 | 79.84 | 73.09 | 79.88 | 73.22 |
| | SxS | NA | NA | 79.61 | 72.56 | 79.73 | 72.82 | 79.82 | 73.04 | 79.87 | 73.17 |
| N=64 | OFMBR | 79.97 | 73.45 | 79.97 | 73.45 | 79.97 | 73.45 | 79.96 | 73.45 | 79.97 | 73.45 |
| | PMBR | **79.90** | **73.07** | **79.98** | **73.30** | **80.02** | **73.44** | **80.02** | **73.47** | **80.00** | **73.47** |
| | NxK | 79.43 | 72.25 | 79.76 | 72.91 | 79.88 | 73.20 | 79.93 | 73.35 | 79.96 | 73.40 |
| | SxS | 79.73 | 72.80 | 79.82 | 73.04 | 79.88 | 73.19 | 79.92 | 73.30 | 79.95 | 73.38 |
| N=128 | FMBR | 79.97 | 73.52 | 79.97 | 73.52 | 79.97 | 73.52 | 79.97 | 73.52 | 79.97 | 73.52 |
| | PMBR | **80.00** | **73.32** | **80.01** | **73.44** | **80.03** | **73.52** | **80.02** | **73.53** | **80.00** | **73.54** |
| | NxK | 79.75 | 72.93 | 79.87 | 73.24 | 79.92 | 73.39 | 79.95 | 73.45 | 79.96 | 73.49 |
| | SxS | 79.87 | 73.17 | 79.91 | 73.29 | 79.94 | 73.37 | 79.96 | 73.44 | 79.97 | 73.49 |
| N=256 | FMBR | 79.96 | 73.60 | 79.96 | 73.60 | 79.96 | 73.60 | 79.96 | 73.60 | 79.96 | 73.60 |
| | PMBR | **80.02** | **73.44** | **80.03** | **73.53** | **80.02** | **73.55** | **80.01** | **73.58** | **80.00** | **73.61** |
| | NxK | 79.86 | 73.28 | 79.90 | 73.42 | 79.94 | 73.49 | 79.95 | 73.52 | 79.96 | 73.56 |
| | SxS | 79.94 | 73.36 | 79.96 | 73.45 | 79.97 | 73.49 | 79.98 | 73.54 | 79.97 | 73.56 |

Table 4: Summary of the average scores of the full EnDe WMT 2022 with N=256 and r=1/16 pairs scored 1000 times using MetricX and COMET22. The MQM scores are limited to 65 examples where all systems disagreed.

|      | COMET22 | MetricX | MQM   |
|------|---------|---------|-------|
| FMBR | 83.33   | 77.15   | 1.169 |
| PMBR | 83.51   | **76.95** | **1.370** |
| NxK  | 83.11   | 76.75   | 1.746 |
| SxS  | **83.59** | 76.79 | 1.566 |

full computational cost on each language pair. The top row comprises the results obtained with the full MBR method (*FMBR*) running on the complete list of $N$=128 candidates, and can be considered as an upper bound for the performance of each approximation method. The number of utility calls for FMBR is $128^2 = 16\,384$. In Figure 3, we plot the data for the for de→en from Table 2. In Table 3, we fix the language pair to (de↔en) and we set $N$ to different values. This simulates the behavior of approximation methods as the candidate list grows. Similar results for (en↔de) are shown in Appendix 6. MQM human evaluation results are summarized in Table 4.

As measuring performance with the same metric we are optimizing for has the risk of overfitting, we mainly focus on COMET22 to assess translation quality. These are the main findings:

**(1) PMBR outperforms all other tested approximation methods**   PMBR outperforms both the $N$x$K$ and $S$x$S$ approximation methods across language pairs, sample sizes and budgets. The gap between the approximation methods closes as the budget increases. Moreover the results in Table **??** show that the same pattern holds when the size of the hypotheses list changes.

**(2) PMBR is competitive to FMBR**   We can reduce the computational cost by up to $r = 1/32$ with PMBR without any loss in translation quality as measured by COMET22. Interestingly, we observe that MetricX scores slightly drop when reducing the budget. As this does not affect the final translation quality as measured by COMET22, we argue that this is a good sign and PMBR acts as some kind of regularization.

**(3) Human Evaluation confirms (1) and (2)**   To verify our findings based on COMET22, we do run a MQM human evaluation with professional translators. Results are summarized in Table 4. The results confirm our previous findings: (1) PMBR is the best approximation method when compared to $N$x$K$ and $S$x$S$, and (2) PMBR is getting close to the performance of FMBR.

## 8   Conclusions

In this paper we have shown the inherent low-rank structure of Minimum Bayes Risk (MBR) score matrices which we leveraged to develop an approximation method for MBR decoding that achieves competitive performance while significantly reducing computational complexity. Our empirical results demonstrate the efficacy of this approach across diverse language pairs and evaluation metrics, suggesting its potential for wider application in machine translation and other natural language generation tasks.

Future research could explore the efficacy of alternative matrix completion algorithms to further enhance the low-rank approximation. In addition, the observed low-rank property could be exploited to inform sampling strategies, potentially leading to more efficient and informative data collection for MBR decoding. Another promising avenue is to investigate the applicability of this work to domains beyond natural language generation tasks.

## 9   Limitations

While we have verified that the MBR matrices are low-rank, we did not conduct an empirical analysis on their coherence. A low-rank matrix is easier to complete if its energy spreads evenly across different coordinates. This property is captured by the notion of coherence (Candes and Recht, 2008).

In this paper, we only run experiments with MetricX as utility function. The computational costs for computing all pairwise utility scores is expensive. However, we showed that the low-rank matrix structure holds for both MetricX and chrF which gives us confidence that PMBR will generalize regardless of the utility function.

Our human evaluation is limited in size because it is costly. With automatic metrics, we can simulate multiple runs of scoring the datasets but this is not feasible with human evaluations. Thus, we put less statistical significance on our human evaluation.

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

# A  Appendix / supplemental material

## A.1  DeEn graphs for all hypothesis size lists

Figures 7, 5 and 6 show the performance of PMBR for different hypothesis list sizes. The number are represented in the main results section in Table 3. PMBR outperforms all the other methods measured by both COMET22 and MetricX.

## A.2  WMT 2022 paragraph level data statistics

We provide the statistics of the dataset used after combining the sentences from the same document in Table 5. The procedure to create paragraph level data is described in details in the data section.

## A.3  Scores for EnDE while varying the hypotheses list size

In Table 6, we summarize all COMET22 and MetricX scores after varying the hypotheses list size. We see similar results as shown in the main results sections. PMBR outperforms all the other approximation methods in most cases.

## A.4  Standard deviations for DeEn and significance test values

In Table 7, we present the standard deviation for values presented in Table 3. We also run a p-value significance test to verify that the gap in performance between PMBR and the other systems is significant.

## A.5  Other matrix completion algorithms

We experimented with running Singular Value Thresholding (SVT) instead of Alternating Least Square (ALS) algorithm to perform the matrix completion. SVT under performed compared to ALS as shown in Figure 8. This behavior might be caused by a mistuning of the SVT hyperparameters, but it highlights the importance of the matrix completion algorithm in the PMBR procedure.

## A.6  Different samples for the hypotheses and pseudo-references lists

In all our experiments, we have used the same set of samples for both hypotheses and pseudo-references which is a common practice for MBR decoding in the NMT use case. In Figure 9, we verify that the matrices are still low rank even when the two set of samples are different. We also run an experiment to benchmark the performance of PMBR in this case. The results are shown in Figure 10

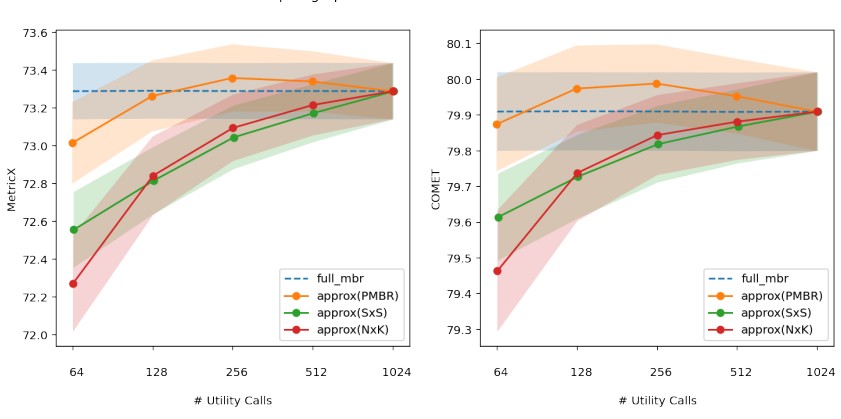

Figure 4: We scored WMT22 DeEn dataset 1000 times for each budget available. Each scoring picks without replacement 32 samples from the 1024 samples available for each sentence. The highlighted area shows the standard deviation of the scores.

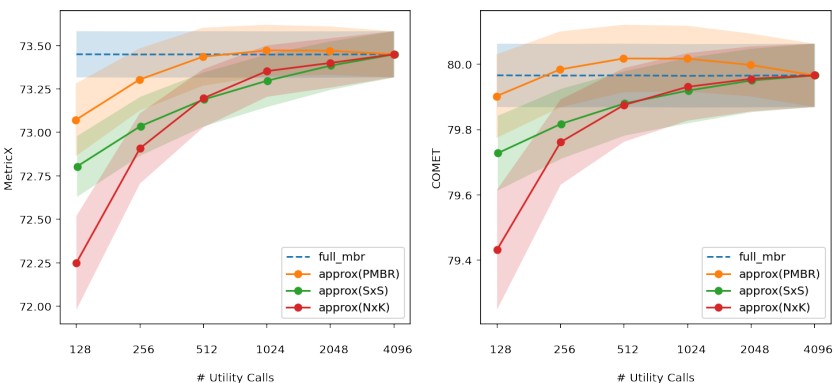

Figure 5: We scored WMT22 DeEn dataset 1000 times for each budget available. Each scoring picks without replacement 64 samples from the 1024 samples available for each sentence. The highlighted area shows the standard deviation of the scores.

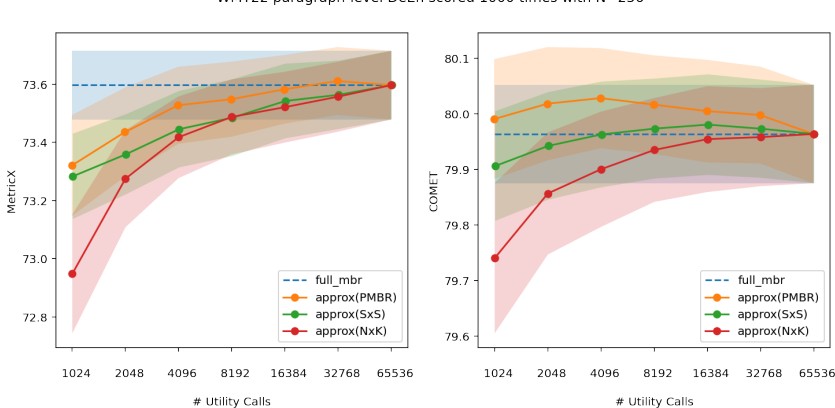

Figure 6: We scored WMT22 DeEn dataset 1000 times for each budget available. Each scoring picks without replacement 256 samples from the 1024 samples available for each sentence. The highlighted area shows the standard deviation of the scores.

### A.7 Compute Resources

We give a high level estimate of the resources to run the experiments:

- Samples Generation: We used around 500 TPUv5 for 5 hours per language pair to generate the samples.
- MetricX pairwise computations: We used around 2000 TPUv4 for 24 hours per language pair to compute all the scores.
- Scoring simulations: These were run on CPUs in parallel on a cluster of 1000 machines. Each setting (budget, hypothesis length) takes around 15 minutes to run.

Table 5: Statistics of the WMT 2022 dataset and its paragraph level transformation.

| LP | | # segments | Avg. #words/segment | Avg. #sent/segment |
|---|---|---|---|---|
| En-De | wmt_22 | 2037 | 16.7 | 1.01 |
| | → paragraph | 219 | 154.6 | 7.99 |
| En-Ru | wmt_22 | 2037 | 16.7 | 1.01 |
| | → paragraph | 219 | 154.6 | 7.99 |
| De-En | wmt_22 | 1984 | 14.6 | 1.01 |
| | → paragraph | 309 | 93.2 | 5.73 |
| Ru-En | wmt_22 | 2016 | 13.6 | 1.01 |
| | → paragraph | 258 | 106.0 | 7.32 |

Table 6: Summary of the average scores of the full EnDe WMT 2022 pairs scored 1000 times using MetricX and COMET as evaluation metrics while varying the size of the hypothesis list

| | budget | 1/32 | | 1/16 | | 1/8 | | 1/4 | | 1/2 | |
|---|---|---|---|---|---|---|---|---|---|---|---|
| | | C | X | C | X | C | X | C | X | C | X |
| N=32 | FMBR | NA | NA | 0.8352 | 0.7641 | 0.8352 | 0.7641 | 0.8352 | 0.7641 | 0.8352 | 0.7641 |
| | PMBR | NA | NA | **0.8294** | **0.7489** | **0.8347** | **0.7571** | **0.8357** | **0.7615** | **0.8356** | **0.7634** |
| | NxK | NA | NA | 0.8072 | 0.7262 | 0.8229 | 0.7445 | 0.8297 | 0.7541 | 0.8335 | 0.7595 |
| | SxS | NA | NA | 0.8255 | 0.7463 | 0.8291 | 0.7525 | 0.8323 | 0.7577 | 0.8339 | 0.7611 |
| N=64 | FMBR | 0.8361 | 0.7679 | 0.8360 | 0.7679 | 0.8360 | 0.7679 | 0.8361 | 0.7679 | 0.8359 | 0.7679 |
| | PMBR | **0.8304** | 0.7504 | **0.8354** | **0.7587** | **0.8366** | **0.7637** | **0.8366** | **0.7662** | **0.8365** | **0.7676** |
| | NxK | 0.8057 | 0.7266 | 0.8226 | 0.7472 | 0.8299 | 0.7577 | 0.8338 | 0.7633 | 0.8354 | 0.7659 |
| | SxS | 0.8293 | **0.7525** | 0.8323 | 0.7578 | 0.8340 | 0.7612 | 0.8352 | 0.7642 | 0.8356 | 0.7662 |
| N=128 | FMBR | 0.8352 | 0.7701 | 0.8352 | 0.7701 | 0.8352 | 0.7702 | 0.8352 | 0.7702 | 0.8352 | 0.7701 |
| | PMBR | **0.8353** | 0.7594 | **0.8363** | **0.7650** | **0.8363** | **0.7681** | **0.8360** | **0.7696** | **0.8356** | **0.7701** |
| | NxK | 0.8218 | 0.7496 | 0.8290 | 0.7599 | 0.8328 | 0.7659 | 0.8345 | 0.7684 | 0.8348 | 0.7693 |
| | SxS | 0.8339 | **0.7612** | 0.8352 | 0.7643 | 0.8357 | 0.7663 | 0.8359 | 0.7679 | 0.8357 | 0.7691 |
| N=256 | FMBR | 0.8332 | 0.7715 | 0.8333 | 0.7715 | 0.8332 | 0.7715 | 0.8332 | 0.7715 | 0.8332 | 0.7715 |
| | PMBR | **0.8354** | 0.7659 | **0.8351** | **0.7695** | **0.8347** | **0.7711** | **0.8341** | **0.7714** | **0.8338** | **0.7717** |
| | NxK | 0.8274 | 0.7617 | 0.8311 | 0.7675 | 0.8326 | 0.7699 | 0.8332 | 0.7710 | 0.8332 | 0.7712 |
| | SxS | 0.8357 | **0.7662** | 0.8359 | 0.7679 | 0.8357 | 0.7692 | 0.8351 | 0.7702 | 0.8344 | 0.7709 |

p-Value significance test for DeEn N=128 for PMBR vs all other systems

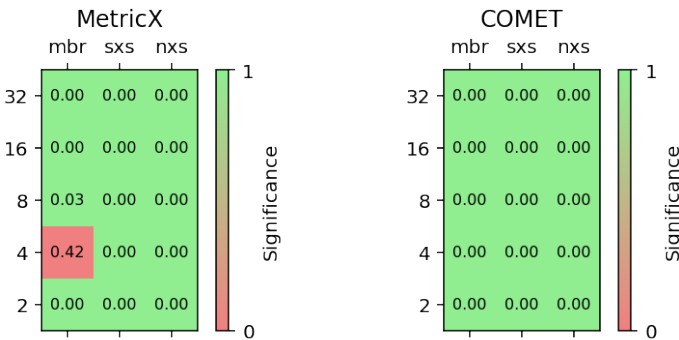

Figure 7: A matrix representing the p-value significance test that checks if PMBR is better than the approximation methods as shown in Table 3. A green boxes means PMBR is significantly better than the other system

Table 7: Summary of the standard deviations of the full DeEn WMT 2022 pairs scored 1000 times using MetricX and COMET22 as evaluation metrics while varying the size of the hypothesis list as shown in Table 3

| | budget | 1/32 | | 1/16 | | 1/8 | | 1/4 | | 1/2 | |
|---|---|---|---|---|---|---|---|---|---|---|---|
| | | C | X | C | X | C | X | C | X | C | X |
| N=32 | FMBR | NA | NA | 0.0014 | 0.0019 | 0.0014 | 0.0018 | 0.0014 | 0.0018 | 0.0014 | 0.0019 |
| | PMBR | NA | NA | 0.0016 | 0.0027 | 0.0015 | 0.0023 | 0.0014 | 0.0022 | 0.0013 | 0.0020 |
| | NxK | NA | NA | 0.0021 | 0.0032 | 0.0017 | 0.0026 | 0.0014 | 0.0022 | 0.0013 | 0.0020 |
| | SxS | NA | NA | 0.0015 | 0.0025 | 0.0015 | 0.0022 | 0.0013 | 0.0021 | 0.0013 | 0.0019 |
| N=64 | FMBR | 0.0012 | 0.0017 | 0.0012 | 0.0017 | 0.0012 | 0.0017 | 0.0012 | 0.0017 | 0.0012 | 0.0017 |
| | PMBR | 0.0016 | 0.0026 | 0.0015 | 0.0022 | 0.0013 | 0.0021 | 0.0013 | 0.0018 | 0.0012 | 0.0017 |
| | NxK | 0.0023 | 0.0034 | 0.0016 | 0.0025 | 0.0014 | 0.0021 | 0.0013 | 0.0018 | 0.0012 | 0.0018 |
| | SxS | 0.0014 | 0.0022 | 0.0013 | 0.0021 | 0.0012 | 0.0019 | 0.0013 | 0.0019 | 0.0012 | 0.0017 |
| N=128 | FMBR | 0.0011 | 0.0015 | 0.0011 | 0.0015 | 0.0011 | 0.0015 | 0.0011 | 0.0015 | 0.0011 | 0.0015 |
| | PMBR | 0.0014 | 0.0023 | 0.0013 | 0.0020 | 0.0012 | 0.0019 | 0.0011 | 0.0016 | 0.0012 | 0.0016 |
| | NxK | 0.0015 | 0.0024 | 0.0014 | 0.0020 | 0.0013 | 0.0018 | 0.0012 | 0.0016 | 0.0011 | 0.0016 |
| | SxS | 0.0013 | 0.0020 | 0.0013 | 0.0019 | 0.0012 | 0.0017 | 0.0012 | 0.0017 | 0.0012 | 0.0017 |
| N=256 | FMBR | 0.0011 | 0.0015 | 0.0011 | 0.0015 | 0.0011 | 0.0015 | 0.0011 | 0.0015 | 0.0011 | 0.0015 |
| | PMBR | 0.0013 | 0.0019 | 0.0011 | 0.0017 | 0.0011 | 0.0016 | 0.0012 | 0.0015 | 0.0011 | 0.0015 |
| | NxK | 0.0014 | 0.0021 | 0.0013 | 0.0018 | 0.0012 | 0.0016 | 0.0012 | 0.0015 | 0.0011 | 0.0015 |
| | SxS | 0.0012 | 0.0017 | 0.0012 | 0.0016 | 0.0011 | 0.0016 | 0.0011 | 0.0016 | 0.0011 | 0.0015 |

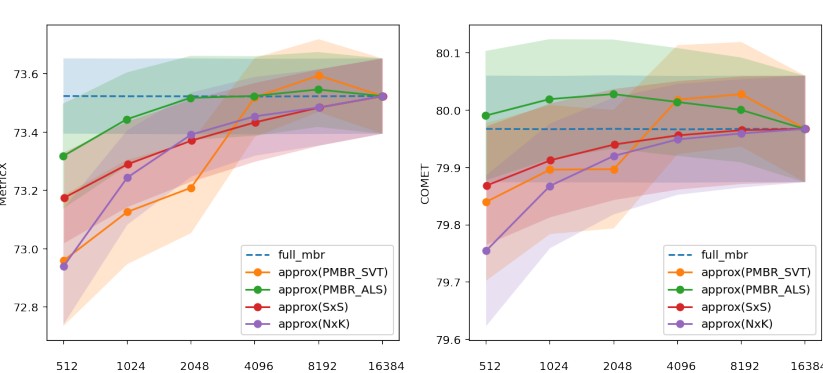

Figure 8: We scored WMT22 DeEn dataset 1000 times for each budget available. Each scoring picks without replacement 128 samples from the 1024 samples available for each sentence. The highlighted area shows the standard deviation of the scores. In this setup, we compare ALS and SVT as matrix completion algorithms.

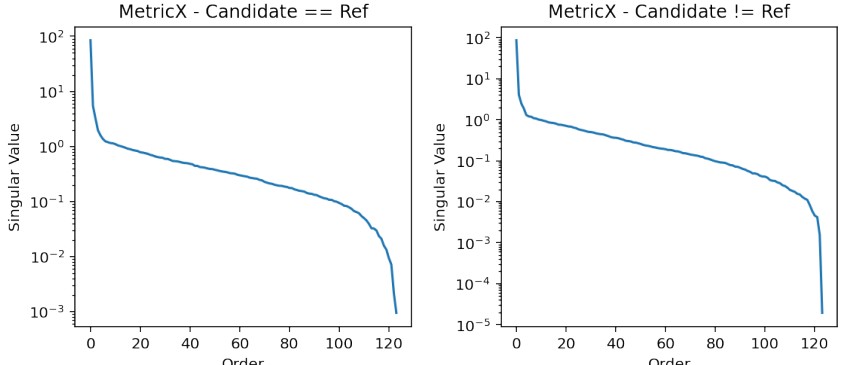

Figure 9: Plot the singular values of an example 124x124 MBR matrix using logscale. The plot on the left shows the case where both the samples for hypotheses and pseudo-references lists, while the right shows the case when they are different. Both plots follow a similar pattern.

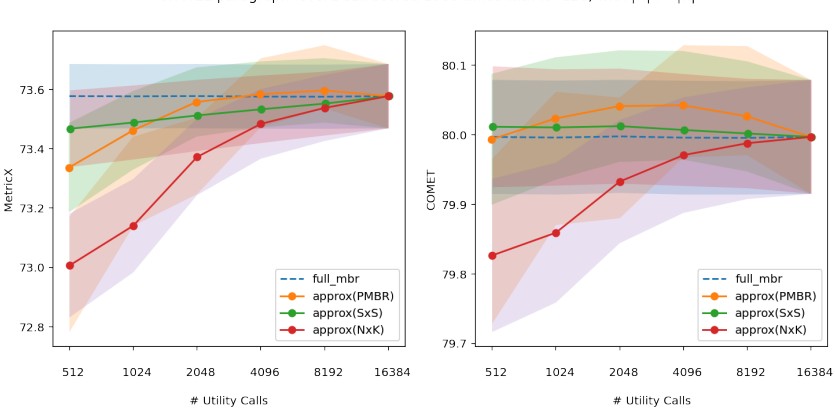

Figure 10: We scored WMT22 DeEn dataset 1000 times for each budget available. Each scoring picks without replacement 128 samples from the 1024 samples available for each sentence. In this setup, we pick different samples for the hypotheses and pseudo-references lists.

