# OpenReview forum: "Efficient Minimum Bayes Risk Decoding using Low-Rank Matrix Completion Algorithms"
_NeurIPS.cc/2024/Conference — NeurIPS 2024 poster_

### Official Review · Reviewer_Xkoo · 2024-07-09

**Soundness:** 3
**Presentation:** 3
**Contribution:** 3
**Rating:** 7
**Confidence:** 5

**Summary:**

Minimum Bayes risk (MBR) decoding is a widely used technique for machine translation (MT) that involves generating $N$ candidate translations that are then scored according to a utility function, typically an automatic reference-based MT metric. In practice, this is done by comparing each candidate in the set to all the other candidates (that are used as pseudo-references), which is computationally very expensive, with a time complexity of O(N^2). This paper focuses on improving the efficiency of MBR decoding by reducing the number of metric computations. Based on the assumption that the MBR utility matrix is low-rank, they score only a subset of the candidate-pseudo-reference pairs and use the ALS algorithm to estimate the missing values.  Their experiments show that their method does not compromise translation quality.

**Strengths:**

- This paper offers a simple solution to a problem that remains unsolved so far (making MBR faster/cheaper) without compromising the final translation quality.
- I believe that framing this as matrix completion problem is a novel approach.
- The experimental results are good and confirmed with human evaluation.

**Weaknesses:**

- The experiments are only performed on two language pairs (EN-DE and EN-RU). Can we expect the same findings for other languages, including lower resource ones? The WMT test sets include other languages, so it shouldn’t be hard to validate this. Also, only PaLM8B is used. It would be good to try other models such as Tower (Alves et al., 2024) with a varying number of samples.
- There is a lot of discussion about related work on making MBR decoding more efficient in Section 2 but the comparison to some of these efficient MBR approaches is missing in the experimental part. Also, it would be good to compare with linear-time cost approaches such as QE-reranking (Fernandes et al., 2022) since this is a clear baseline.
- See my questions below for other things.

Tower: An Open Multilingual Large Language Model for Translation-Related Tasks (Alves et al., 2024)

Quality-Aware Decoding for Neural Machine Translation (Fernandes et al., 2022)

**Questions:**

Comments and questions:
- Although the definition of MBR matrix is presented in Section 4, it would be beneficial to briefly explain this a bit better earlier in the paper (e.g., in the paragraph in L44-54). It may not be straightforward to all readers to understand where this matrix comes from and why it can be assumed to be low-rank.
- Where is Figure A.6 and Table 4.2, mentioned in L155? This is pointing to the sections… After a closer look, Table 1 and Figure 5 seem to be the ones that you are referring to.
- What happens if you fill in the missing values in the MBR matrix with random numbers instead of using ALS? I believe this is an important baseline.
- The results in the tables are averaged over 1000 trials. Can you please provide standard deviations too?

Minor comments:
- Fix citation style in L84 (no parenthesis) and L86 (missing whitespace).
- Missing punctuation in Eq. 2.
- Missing whitespace in L202.

**Limitations:**

Yes.

---

> ### Author Rebuttal · Authors · 2024-08-06
>
> Thank you for the detailed feedback.
>
> We limited the paper to 4 different language pairs because precomputing the full 1024x1024 utility metrics is very expensive (It requires around 2000 TPUs for a full day per language pair). We definitely agree that performing a survey across all the efficient MBR methods will be beneficial for the community and we are interested in running this for future research. We will point this out in the limitations section of our paper.
>
> We address your questions in order:
> 1) Thanks for the suggestion. We agree that the MBR matrix definition is not straightforward and that’s why we included a dedicated section to introduce it. We will update paragraph 4 to give a high level overview and point out to section 4 for the full definition.
> 2) Table 1 and Figure 5 are the correct artifacts. Figure 5 was in the main paper but we moved it to the appendix when we were trimming down the paper to fit in the page limit. We will correct the labels and move figure 5 back to the main paper.
> 3) We believe that this method should be in practice very similar to the NxS if we assume that on average the random values will add the same value to each row.
> 4) We attach table 7 that contains all the standard deviations from table 3 shown in our submission. All methods showed similar trends in standard deviation values. We will add the numbers to the paper.
>
> Thank you again for your review and thanks for catching all the minor style issues. We will make the adjustments accordingly in the paper.

---

> > ### Comment · Reviewer_Xkoo · 2024-08-08
> >
> > Thank you for answering my questions. After reading the rebuttal and the other reviews, I think that incorporating some of this discussion in the updated version of the paper, including the limitations you mentioned above, is a good idea.

---

### Official Review · Reviewer_PUED · 2024-07-11

**Soundness:** 3
**Presentation:** 3
**Contribution:** 3
**Rating:** 6
**Confidence:** 3

**Summary:**

- this paper uses alternating least square method for matrix completion when only a small part of the utility matrix in MBR decoding is calculated to decrease the computational cost for full MBR decoding and at the same time guarantee the translation quality
- the authors first justify the use of their method by showing that the utility matrix in MBR decoding is indeed low rank
- by conducting experiments on WMT dataset and four translation directions, they show that PMBR achieved the best COMET22 score and MetricX on most of the settings

**Strengths:**

- the method they propose is a simple yet effective one to shrink the computational cost for MBR decoding in NMT task
- Across all the experimental settings the metric score is indeed better than other baseline methods while keeping the cost lower

**Weaknesses:**

- ALS is one method for matrix completion, ablations should be done using other matrix completion methods to see if the cost can be further reduced by using less examples
- using matrix completion method to lower the cost for computation intensive tasks itself is not a novel approach
- PMBR is compared against other MBR decoding related methods but not compared against e.g. naive beam search

**Questions:**

- there are other matrix completion methods that also focus on picking the most representative matrix entries for completion. Why did you not compare against those methods?

**Limitations:**

- yes the authors addressed the limitations
- and there is no potential societal impact

---

> ### Author Rebuttal · Authors · 2024-08-06
>
> Thanks for the insightful review.
>
> We mainly focused on ALS because of its simple implementation and efficiency. We agree that optimizing the matrix completion algorithm further could potentially improve the performance. As you suggested, using an adaptive sampling[1] approach to pick the most representative matrix entries is a promising future direction but we chose the simplest approach to make it easy for others to adopt our work.
>
> We run ablations with SVT(singular value thresholding) on one experiment (Lp= DeEn, N=128) which is another elementary matrix completion algorithm. On average, ALS outperforms SVT in particular for tighter budgets [fig1]. This emphasizes that investigating the performance of the matrix completion algorithm is certainly an interesting future direction. We will add this comparison and a discussion in the paper.
>
> We added results for greedy decoding [fig2]  (metricx:70.19, comet=76.74) and the results are in line with previous work[2]. Greedy decoding falls behind with almost 3 points on both Comet and MetricX. We will add these results to the paper.
>
> Thanks again for this great and insightful review.
>
> [1] https://arxiv.org/abs/1304.4672
>
> [2] https://arxiv.org/abs/2111.09388

---

> > ### Comment · Reviewer_PUED · 2024-08-13
> >
> > Thank you for your response and the additional experimental results with SVT and the greedy decoding baseline. i agree that ALS  is a simple yet efficient matrix completion algorithm for this use case. I will raise my score to 6.

---

### Official Review · Reviewer_GHFT · 2024-07-12

**Soundness:** 3
**Presentation:** 3
**Contribution:** 3
**Rating:** 7
**Confidence:** 4

**Summary:**

This work proposes an approximate method for minimum Bayes risk (MBR) decoding by employing matrix completion. Basic idea is to fill-in only a fraction of the score matrix to compute MBR scores, and to leverage alternating least squares (ALS) algorithm to estimate the empty slots assuming that the utility score matrix is low rank. This work empirically show that the utility score matrix is actually low rank for MetricX and chrF under some language directions. Empirical results also show that the proposed approximate MBR decoding is a better approximation method when compared with other approximation method, e.g., shrinking the dimension of pseudo-reference list size and sampling-based method.

**Strengths:**

- The proposed method is interesting in that it leverages the low-rank matrix completion to estimate the full utility score matrix for MBR decoding assuming that utility score matrices are low rank. The singular values of MT metrics empirically show that the low-rank property holds for some language directions. I think the finding somewhat agrees withe the matrix completion for human assessment [1].
- Empirical results show that the proposed method is better than other approximation methods even with only a small number of samples, and achieving comparable performance to the non-approximation method even the budget for 1/4. Human judgement also shows that the proposed method is closer to the non-approximation method.

[1] https://aclanthology.org/P13-2025/

**Weaknesses:**

- This work needs further analysis on the utility score matrices after convergence since there exist some gaps when compared with the non-approximation method under the budget of 1/32. In particular, the true scores filled by the metric might be changed after completion, and I'm curious how that will impact MBR decoding. Also, it would be good to analyze the deltas of the completed scores to see how that will impact the end accuracies.
- In addition to random sapling to determine the position to fill-in by the true score, I'd like to see a more controlled setting in which the positions are skewed, in which some columns or rows are completely empty and thus complemented by ALS, and how that will impact the estimated scores and the final MBR decoding.

**Questions:**

Questions:
- It is not clear how many iterations are needed for the convergence in general. Also, it would be good to show the average differences and variances of the score matrices by the approximated methods, i.e., prediction performance for the missing values, to quantify the impact of the approximation.

Suggestions:
- I think Figure 1 is not mentioned anywhere in the paper, and please describe what is indicated by the plots. Also, do not use a bitmap format but use a vector format for clarity.
- line 155: Figure A.6 and Table 4.2 (?)  -> please check the references.
- Better to run statistical significance tests on the numbers in Table 2 and 3 to make sure the differences are statistically significant or not.

**Limitations:**

This paper has discussion on the limitations, e.g., measuring only MetricX, but this work is actually running experiments for COMET as a utility function.

---

> ### Author Rebuttal · Authors · 2024-08-06
>
> Thanks for the insightful review.
>
> We agree that a further analysis on the convergence of the matrix completion algorithm will help us understand better the behavior of our approach and potentially optimize it further. We think this is an interesting future direction of research and we we will point this out in the paper.
>
> The number of iterations needed for convergence is a hyperparameter that we tune. In general, the required number of steps is less than 15. We included a graph [fig 4] that shows the true loss and training loss during the approximation (true_loss = ((M-A)**2), train_loss = mean(M[omega]-A)**2) with M : original matrix, Omega: sampling mask and A : the approximation at every step). We will include this graph in the paper.
>
> A more sophisticated sampling is an interesting approach to optimize our method further. We definitely want to explore this behavior in future research, one promising approach is adaptive sampling[1] where we pick the samples that give us the most information about the matrix.
>
> We run a significance test for PMBR against all other systems for the values shown in (Table 3, row 3) in the paper. We see that the gap between PMBR and all the other systems is significant except for one setting [fig 5]. (Note, that greens denote the significance test and not that PMBR is better). We will include these numbers in the paper.
>
> Thank you again for your review and thanks for pointing out the missing references. We will make the adjustments accordingly in the paper.
>
>
> [1] https://arxiv.org/abs/1304.4672

---

> > ### Comment · Reviewer_GHFT · 2024-08-09
> >
> > Thank you for extra experiments.
> >
> > I feel this work is solid enough with meaningful comparisons.

---

### Official Review · Reviewer_D2EM · 2024-07-13

**Soundness:** 3
**Presentation:** 2
**Contribution:** 4
**Rating:** 6
**Confidence:** 3

**Summary:**

The paper proposes yet another method for speeding up the utility computation of MBR decoding. The algorithm exploits the structure of the problem that the utility matrix tends to be lower rank as the utility is kind of a similarity metric. The experiments are compared against the standard MBR and show that it achieves a good speedup over the standard MBR.

**Strengths:**

- The proposed algorithm is interesting. Prior works do not consider the structure of the utility matrix and rather seek to optimize the computation via hyperparameter tuning or to optimize the worst-case complexity.
- The observation that the utility matrix is low rank is interesting and is a contribution to the community.
- Human evaluation is nice to have.

**Weaknesses:**

The paper brings an interesting new idea to MBR decoding. However, their contribution is claimed ambiguously.

- The experiment does not compare any of the recent algorithms on efficient MBR decoding: Confidence-based pruning (Cheng and Vlachos 2023), Adaptive MBR (Jinnai and Ariu 2024), Linear time MBR (Vamvas and Sennrich 2024), and Centroid-based MBR (Deguchi et al. 2024). Thus, the paper does not show evidence that their method is state-of-the-art with respect to the empirical performance, which will not be a reason for rejection.
- Empirical evaluation of concurrent work is too much to ask, but I would say that they should be explained fairly. Vamvas and Sennrich (2024) are mentioned that “the utility metric needs to fulfill certain conditions to be applicable.” (Line 77). This is true but I would say that “certain conditions” should be explained clearly in this paper, because the proposed method is only effective when these “certain conditions” are not satisfied, if I understand it correctly. Also note that the proposed method also exploits the structure of the utility metric that results in a low-rank matrix, which may not be universally true. The proposed method also requires that certain conditions are satisfied. I believe this will not be a reason to reject, only if this limitation is clarified in the paper.

**Questions:**

- Figure 5 shows that the utility matrices are likely to be rank-1. My understanding is that in theory it requires a single comparison per each candidate is enough to estimate the quality of the candidate, assuming that we can ignore the noise. I was thinking if this observation would rather motivate the Linear time MBR (Vamvas and Sennrich 2024) as they use a single comparison against the mean of the references, which we expect to have the least noise.
- Although the rank of the matrix is estimated to be 1, the ranks of the matrices are set to 5 to 15 (Line 224) in the experiment. Why is it better to set it larger than 1? I’m assuming r in Line 224 refers to the rank of the matrix. r is used as the rank and also the reduction ratio in Algorithm 1.
- What is the definition of O(millions) and O(hundreds)?  (Line 171)
- Does the method require that candidates and references are the same set of samples? I see several papers in MBR using different sets of samples (Cheng and Vlachos 2023; Ohashi+2024) so it would be better if it works with both cases. I don’t see it as a reason to reject it, but I think it needs to be clearly mentioned that the proposed method can’t use a different set of samples, if so.
- It would be nice to have a comparison of the overall walltime, including the time for sample generation to show in the end how much it improves the speed of the NMT system.

Typos
- Line 51: effectiveness of out method → effectiveness of our method
- Line 91: a NMT model → an NMT model
- Line 155: Figure A.6 and Table 4.2 → I guess Figure 5 and Table 1
- Line 196: SPM tokens → no definition of SPM
- Line 197: Note:
- Line 414 Algorithm 2: There are N and n that I believe they both refer to the same parameter.
- Line 431: The sentence is incomplete.
- Line 513, 522: Doesn’t MQM involve human subjects?

**Limitations:**

I think the paper does not explicitly explain the critical limitations of the method.

---

> ### Author Rebuttal · Authors · 2024-08-06
>
> Thanks for the insightful review.
>
> We definitely agree that performing a survey across all the efficient MBR methods will be beneficial for the community and we are interested in running this for future research. We will point this out in the limitations section.
>
> We address your questions in order:
> * This is a great observation. During our work, we were trying to theorize what this first singular vector semantically means and as you point out one comparison against this vector should be enough to tell us the quality of the candidate. We think it’s interesting to compare this vector against the reference aggregation embedding (Vamvas and Sennrich 2024) proposed in their work. One key advantage of their approach is the linear-time complexity, but they require the utility metric to produce an intermediate sentence embedding. On the other hand, our approach is slower and requires that the MBR matrices have a low-rank structure. Exploring how these two methods overlap is definitely an interesting direction for future research. We will include this insight in the discussion section of the paper and we will emphasize the low-rank condition in the limitation section.
> * Correct, there are two ranks. The true rank r of the matrix and the rank r’ which is a hyperparameter specific to ALS. We’ve run a hyperparamter sweep and we found that using r=5 worked best to minimize the training loss.
> We do not see any reason that this method requires candidates and references to be the same. Looking at the singular values structure we see that both settings exhibit similar behaviors [fig .3]. We promise to run more experiments to validate this setting and report in the final paper.
> * Regarding the time complexity, the paper states O(millions) and O(hundreds) matrix multiplications but `floating point operations` is more accurate. We will update the paper. The O(millions) is used as an abbreviation for 10^6 order of magnitude, this is an abuse of the big O notation and we will update the paper accordingly too. We did not compute or report actual wall clock time because the experiments are run on a mixture of hardware and configurations (different TPUs, different CPUs) and the walltime will heavily depend on this. However, measuring the number of MBR computations as shown in our paper is in our opinion a fairer and more straightforward measure.
>
> Thank you again for your review and thanks for catching all typos. We will make the adjustments accordingly in the paper.

---

> > ### Comment · Reviewer_D2EM · 2024-08-09
> >
> > Thank you very much for the clarification.
> >
> > My position is unchanged. The paper brings an interesting new idea to the field of MBR decoding. Yet, it is not compared against the recent algorithms which makes it unclear if it is on par with the existing work in benchmarks. I think the comparison should not be future work and this is the paper that should conduct the comparison if it is to be claimed as a state-of-the-art method.

---

### Author Rebuttal · Authors · 2024-08-06

Thank you all for the reviews. We attach a pdf that includes more results and details that were requested. The plots in the pdf are referenced in the individual replies with [fig #number] format.

---

### Decision · Program_Chairs · 2024-09-25

**Decision:**

Accept (poster)

**Comment:**

This paper proposes a method to improve the efficiency of MBR decoding by reducing the number of metric computations. They show that the utility matrices for MBR decoding are indeed low-rank,  and they score only a subset of the candidate-pseudo-reference pairs and use the ALS algorithm to estimate the missing values to approximate the full MBR matrix. Experiments  on WMT dataset show that this method  achieves a good speedup over vanilla MBR and does not compromise translation quality.

Some reviewers raised concerns regarding to the comparison with the recent algorithms on efficient MBR decoding, such as Confidence-based pruning, Linear time MBR, Centroid-based MBR,  QE-reranking and the naive beam search, to make it more clear if this work is on par with the existing work. The authors provided some clarifications and limitations in their rebuttal with some extra results. Finally, all reviewers feel positive to the work. My comment is that this work brings an interesting new idea and makes a good contribution to the field of MBR decoding, with minor issues that can be solved in the revision. Therefore, I recommend an "acceptance".